# Obstructive Sleep Apnea as a Risk Factor of Insulin Resistance in Nondiabetic Adults

**DOI:** 10.3390/life11010050

**Published:** 2021-01-13

**Authors:** Monika Michalek-Zrabkowska, Piotr Macek, Helena Martynowicz, Pawel Gac, Grzegorz Mazur, Magda Grzeda, Rafal Poreba

**Affiliations:** 1Department of Internal Medicine, Occupational Diseases, Hypertension and Clinical Oncology, Wroclaw Medical University, 213 Borowska St., 50-556 Wroclaw, Poland; monika.michalek@student.umed.wroc.pl (M.M.-Z.); macekpiotr@op.pl (P.M.); helenamar@poczta.onet.pl (H.M.); grzegorz.mazur@umed.wroc.pl (G.M.); sogood@poczta.onet.pl (R.P.); 2Department of Hygiene, Wroclaw Medical University, 7 Mikulicza-Radeckiego St., 50-345 Wroclaw, Poland; pawelgac@interia.pl

**Keywords:** obstructive sleep apnea, insulin resistance, nondiabetic patients

## Abstract

Objective: The aim of this research was to assess the relationship between prevalence and severity of obstructive sleep apnea (OSA) and insulin resistance among patients with increased risk of OSA without diabetes mellitus. Method and materials: our study group involved 102 individuals with suspected OSA, mean age 53.02 ± 12.37 years. Data on medical history, medication usage, sleep habits, sleep quality and daytime sleepiness, were obtained using questionnaires. All patients underwent standardized full night polysomnography. Serum fasting insulin and glucose concentration were analyzed, the homeostatic model assessment-insulin resistance (HOMA-IR) index was calculated. Results: polysomnographic study indicated that in the group with OSA mean values of apnea–hypopnea index (AHI), oxygen desaturation index (ODI), duration of SpO_2_ < 90% and average desaturation drop were significantly higher compared to the group without OSA, while the minimum SpO2 was significantly lower. The carbohydrate metabolism parameters did not differ within those groups. Significantly higher fasting insulin concentration and HOMA-IR index were found in the group with AHI ≥ 15 compared to the group with AHI < 15 and in the group with AHI ≥ 30 compared to the group with AHI < 30. Higher AHI and ODI were independent risk factors for higher fasting insulin concentration and higher HOMA-IR index. Increased duration of SpO2 < 90% was an independent risk factor for higher fasting glucose concentration. Conclusions: Individuals with moderate to severe OSA without diabetes mellitus had a higher prevalence of insulin resistance.

## 1. Introduction

Obstructive sleep apnea (OSA) is a sleep-related breathing disorder characterized by collapsing of the upper airway during sleep. In individuals with this disorder, obstructed pharyngeal airway results in total respiratory cessation (obstructive apnea) or reduction of ventilation (hypopnea) and snoring. Magnified respiratory effort results in arousal due to reopen the airway and continue breathing [1]. The recurrent hypoxia and hypercapnia cause increased oxidative stress due to inadequate arterial blood oxygenation and metabolic demand contributing to cardiovascular and cerebrovascular morbidity and mortality. This dependency was demonstrated in many epidemiologic, cross-sectional and prospective pieces of research such as the sleep heart health study [2], the Wisconsin sleep cohort study [3], the HypnoLaus study [4].

The prevalence of moderate to severe OSA in the United States in adults is estimated to be between 10% to 17%, among men aged 30–70 y and about 3% to 9% in women between 30 and 70 y.o [5]. In another cohort study of 2121 people who underwent polysomnography-HypnoLatus conducted in Europe, Switzerland, the prevalence of moderate to severe OSA was even higher—about 23,4% in women and 49.7% in men, mean age of participants was roughly 60 y [4]. Compared to previous population-based studies [6,7], the prevalence of sleep-disordered breathing is still rising [3,5].

Obstructive sleep apnea is associated with many comorbidities: it is an independent risk factor for incident coronary heart disease, it is also a significant predictor of incident heart failure [2], and it is an independent risk factor of stroke and death from any cause [8,9]. OSA is also linked with depression and cognitive impairment [10], may promote or exacerbate hypertension [11,12], arrhythmias [13,14] and systemic inflammation [15]. Clinical data also indicate that increase in weight causes an increase in the odds ratio of developing OSA [16]. Moreover, OSA was also independently associated with an increased prevalence of metabolic syndrome [17], type 2 diabetes mellitus [18,19,20,21,22], obesity [22,23,24], glucose intolerance and insulin resistance (IR) [21,25]. The article by Bikov et al. brought some information about the background of the problem in the Central European population showing the raw prevalence of OSA comorbidities in Romanian and Hungarian populations. For diabetes, prevalence estimated as follows: about 25% for the Hungarian cohort and about 20% for the Romanian cohort [26].

The association between obstructive sleep apnea and diabetes has been discussed by a great number of authors in literature. However, the association between OSA and insulin resistance in nondiabetic patients is still insufficiently explored.

Diabetes mellitus is considered a crucial risk factor for sleep-related breathing disorders due to the obesity epidemic [18,19,20]. OSA may also have an influence on disturbed glycemic control and adipose tissue malfunction. Insulin resistance and glucose intolerance are common in subjects with OSA; however, the evidence linking main parameters measured by polysomnography and self-reported sleepiness with IR, glucose intolerance and fasting glucose serum concentration needs verification.

The aim of this research was to give information about links between the prevalence and severity of OSA and insulin resistance in nondiabetic individuals among patients suspected of having OSA without diabetes mellitus. We decided to examine the association between PSG parameters, sleep stages, daytime sleepiness scored with ESS and insulin resistance.

## 2. Materials and Methods

### 2.1. Study Sample

The study was conducted in the Sleep Laboratory in the Department of Internal Medicine, Occupational Diseases, Hypertension and Clinical Oncology at Wroclaw Medical University, Poland. We enrolled all consecutive patients admitted to our Department for polysomnography due to OSA diagnostics. After taking medical interviews and reading the medical history of patients, we have excluded individuals with diabetes mellitus, taking hypoglycemic medications, or those who were unable to give informed consent. Individuals with incident diabetes mellitus (with fasting glucose concentration of ≥126 mg/dL) were excluded from the study. Objects with severe pulmonary disorders, e.g., asthma or chronic obstructive pulmonary disease exacerbation, active malignancy, respiratory insufficiency or active inflammation, were also excluded.

Our final study group estimated 102 individuals with suspected OSA, 71 men and 31 women, mean age 53.02 ± 12.37 years; 64.7% with reported arterial hypertension. Self-reported daytime sleepiness assessed with ESS was scored at 9.09 points SD ± 5.13.

### 2.2. Clinical Assessment

First, patients were distributed within 2 categories: with OSA (group A) and without OSA (group B). Table 1 summarizes data on the demographic and polysomnographic variables of the study sample. Table 2 refers to glucose metabolism according to OSA criteria.

Data on medical history, medication use, self-reported sleep habits and sleep quality and daytime sleepiness were obtained using questionnaires. In this research, for assessing daytime sleepiness, we used the Epworth sleepiness scale.

According to carbohydrate metabolism, in the next step of the analysis, we distributed all patients within 6 groups depending on OSA severity and reported sleepiness: group C with AHI rather than or equal to 15 and group D with apnea–hypopnea index (AHI) estimated below 15, group E with AHI greater or equal to 30 and group F with AHI below 30 (Table 3). Table 4 is referred to glucose metabolism according to Epworth sleepiness scale criteria; individuals were divided into 2 groups as it is shown in the table: group G with ESS rather than or equal to 11 points and group H with ESS points below 11.

### 2.3. Polysomnography

All patients underwent standardized full night polysomnography using NOX-A1 (Nox Medical, Reykjavik, Iceland). Polysomnograms were scored in 30 s epochs. Sleep stages were classified according to standard sleep criteria by the American Academy of Sleep Medicine Task Force. Polysomnography outcome variables included sleep latency, REM latency, total sleep time, sleep efficiency (%), the ratio of N1, N2, N3 sleep stages and the stage of REM. Abnormal respiratory events were scored from the standardized airflow signal: apneas were defined as the absence of airflow for ≥10 s, hypopneas were scored as a reduction in the amplitude of breathing by ≥30% for ≥10 s with a ≥3% decline in blood oxygen saturation or followed by arousal.

### 2.4. Metabolic Assessment

A blood sample was obtained by venipuncture after 12 h overnight fast. Serum insulin and glucose concentrations were analyzed by the Hospital’s Main Laboratory. On the basis of fasting plasma insulin and glucose levels, the HOMA-IR index was calculated using the formula above: fasting glucose (G) (mmol/liter) and fasting insulin (I) (µU/liter) values divided by the constant 22.5: HOMA = (G × I)/22.5. The HOMA index correlates well with insulin resistance.

### 2.5. Statistical Analysis

Statistical analyses were performed using the statistical package “Dell Statistica 13.1”. (Dell Inc., Round Rock, TX, USA). The distribution of variables was checked by Lilliefors and W-Shapiro–Wilk tests. The *t-*test was used for the independent quantitative variables with a normal distribution. For variables with distribution other than normal, the Mann–Whitney *U-*test was used for quantitative independent variables. For independent qualitative variables, the quadrate-square test of the highest reliability was used. Correlation and regression analysis were performed to determine the relationships between the variables studied. The parameters of the model obtained in the regression analysis were estimated using the least-squares method. The results on the level of *p* < 0.05 were assumed to be statistically significant.

## 3. Results

A total of *n* = 102 patients were included in our research. Among nondiabetic individuals from our study, the majority n= 85 (83.3%) had OSA, 64 men (75.3%) and 21 (24.7%) women. This study revealed that the difference between group A and B in the presence of arterial hypertension and age of individuals was not significant. Polysomnographic, anthropometric and laboratory findings are shown in Table 1. Statistically significant divergences between groups A and B (with and without OSA) are also demonstrated in Table 1.

To provide information about carbohydrate metabolism in apneic and non-apneic patients, we analyzed fasting glucose insulin levels, HOMA index, impaired fasting glucose (IFG) and impaired glucose tolerance (IGT). The difference remained statistically nonsignificant (Table 2). Because we did not collect blood glucose after an oral glucose tolerance test due to short hospitality time, data on IGT and IFG were poor (7/6.9% and 15/14.7%, respectively), but the differences were not statistically significant.

The next analysis compared carbohydrate metabolism in patients divided on the basis of OSA severity. There were statistically significant differences between study groups C (with moderate to severe OA) and D (the rest of the study group) according to insulin concentration and HOMA index (*p* < 0.05). There were no differences in glucose concentration, IGT and IFG. According to severe OSA (AHI greater or equal to 30, group E) and the rest of the study group (AHI less than 30, group H), the statistically significant differences were found in insulin concentration and HOMA index (*p* < 0.01)

The effect of sleepiness assessed with ESS on carbohydrate metabolism was also analyzed. Groups with ESS greater than or equal to 11 (group G) and ESS less than 11 (group H) were compared. There were no statistically significant differences in glucose and insulin concentration, HOMA index and IGT, IGF (Table 4).

Subsequently, correlation analysis was performed. The concentration of glucose, insulin, HOMA index correlated with various polysomnographic parameters are shown in Table 5.

Regression analysis was performed in the study group, and independent predictors of increased glucose concentration (Table 6), insulin concentration (Table 7) and HOMA index (Table 8) were determined. Independent predictors of higher glucose concentration were age and duration of SpO_2_ < 90%. In the next step, ODI, AHI and age were determined as independent predictors of higher insulin concentration. AHI, ODI and BMI were independent factors for higher HOMA index values. For the estimation of the model with glucose R^2^= 0.275, for the model with insulin R^2^ = 0.323, and for the model with HOMA-IR R^2^ = 0.284.

## 4. Discussion

The aim of the current study was to assess the effect of obstructive sleep apnea on insulin resistance in nondiabetic patients. Patients without severe pulmonary disturbances were admitted to the study. Obstructive sleep apnea was diagnosed using polysomnography. There was no significant difference in carbohydrate metabolism between patients with OSA and non-OSA. In the next analysis, we considered the impact of OSA severity based on AHI and the impact of sleepiness based on the Epworth sleepiness scale. In the first analysis, we have obtained that OSA severity had a significant impact on carbohydrate metabolism. Patients with AHI greater than or equal to 15 and with AHI greater than or equal to 30 had significantly higher insulin concentration and HOMA index. Correlation analysis also confirmed that these parameters were positively correlated with OSA severity. A significant positive correlation between glucose concentration and the duration of SpO_2_ < 90% was also obtained. According to sleepiness, the analysis showed no association between subjective scale ESS and carbohydrate metabolism. In our study, we also determined independent predictors of higher insulin and glucose concentration and HOMA index. This study revealed that higher AHI values positively correlated with increased insulin concentration and the HOMA index but did not correlate with serum glucose concentration.

The topic of insulin resistance in patients with obstructive sleep apnea has already been raised in the literature. There were some publications on this subject in databases. Results of the current study go beyond previous reports showing that OSA is independently associated with insulin resistance [27,28]. A previous study by Liu et al. attended by 4152 participants showed that obstructive apnea and a decreased level of lipoproteins were correlated with insulin resistance. Authors showed that the severity of OSA, expressed using the AHI value, had a significant impact on insulin resistance assessed by the HOMA index [29]. These findings were in line with our results. However, previous studies did not determine the effect of increased sleepiness on insulin resistance.

The next study by Archontogeorgis et al. analyzed insulin resistance in patients with obstructive sleep apnea. In this study, patients with obstructive sleep apnea were divided into two groups based on the HOMA index: greater than or equal to 2 and HOMA index less than 2. Correspondingly to our results, patients with higher HOMA index had significantly higher AHI and duration of desaturation. Unlike our study, there were statistically significant differences in glucose levels between the two groups. Researchers also demonstrated that sleepiness severity measured using ESS did not affect insulin resistance. The apparent advantage of this study was a large group of 446 individuals [30]. The current study demonstrated that increased duration of desaturation time below 90% is an independent risk factor for higher fasting glucose levels. This also has been explored in a prior study by Sulit et al. Individuals with 2% and more duration of saturation less than 90% had 2.33 times odds ratio (95% CI 1.38, 3.94) of impaired glucose tolerance [31]. The availability of oral glucose tolerance tests was the robust advantage of that study.

The strengths of our study were that full diagnostics of respiratory disorders were carried out using polysomnography, giving the opportunity to show many parameters that can change, especially in patients with obstructive sleep apnea. In addition, a large group of patients took part in our study. These patients were relatively young and heterogeneous, which made it possible to eliminate the influence of factors that develop with age and which may affect insulin resistance. The weakness of our study was the short time which patients spent in the hospital and the inability to collect all important medical data and analyze the impact of treatment on their insulin resistance. Therefore, in the next step, it would be reasonable to analyze the impact of the treatment of patients with OSA on their insulin resistance. A significant limitation of the study is the lack of data on AHI measured separately in REM and NREM sleep stages. The next apparent limitation is the lack of data on diet, alcohol intake and smoking. However, we will consider surveying patients with extensive questionnaires when designing subsequent research. This provides a good starting point for discussion and further research.

## 5. Conclusions

1. Individuals with moderate to severe obstructive sleep apnea without diabetes mellitus had a higher prevalence of insulin resistance estimated with the HOMA-IR index;

2. This study revealed that there is no evidence that the evaluation of daytime sleepiness using the Epworth sleepiness scale is a predicting factor for developing IR;

3. Increased AHI and ODI were independent risk factors for increased fasting insulin concentration and higher HOMA index;

4. Increased duration of desaturation time below 90% is an independent risk factor for higher fasting glucose concentration.

## Figures and Tables

**Table 1 life-11-00050-t001:** Baseline characteristics and polysomnographic parameters of studied population stratified by the presence of obstructive sleep apnea (OSA).

	Whole Study Group *n* = 102	OSA (Group A) *n* = 85	Without OSA (Group B) *n* = 17	*p* A–B
Men	71/69.6	64/75.3	7/41.2	<0.01
Women	31/30.4	21/24.7	10/58.8	<0.01
Age (years)	53.02 ± 12.37	53.38 ± 12.45	51.24 ± 12.16	ns
Height (m)	1.74 ± 0.09	1.75 ± 0.09	1.69 ± 0.10	ns
Body mass (kg)	95.56 ± 27.74	95.97 ± 28.69	90.50 ± 11.70	ns
BMI (kg/m^2^)	30.50 ± 6.29	30.69 ± 6.69	29.53 ± 3.74	ns
Arterial hypertension	66/64.7	55/64.7	11/64.7	ns
Total cholesterol (mg/dL)	205.91 ± 47.07	208.94 ± 47.89	190.94 ± 40.78	ns
HDL cholesterol (mg/dL)	48.70 ± 12.53	48.25 ± 11.98	50.94 ± 15.17	ns
LDL cholesterol (mg/dL)	123.21 ± 38.95	124.83 ± 39.81	115.41 ± 34.50	ns
Triglycerides (mg/dL)	183.58 ± 88.61	189.95 ± 92.44	152.47 ± 59.60	ns
AHI (n/h)	25.78 ± 24.84	30.49 ± 24.64	2.23 ± 1.49	<0.001
ODI (n/h)	24.99 ± 23.25	29.31 ± 23.14	3.38 ± 2.42	<0.001
TST (min)	388.46 ± 73.17	392.52 ± 71.70	368.18 ± 79.23	ns
SL (min)	27.28 ± 33.05	25.82 ± 34.05	34.62 ± 27.23	ns
REML (min)	82.63 ± 62.90	80.39 ± 60.04	93.85 ± 76.73	ns
WASO (min)	62.32 ± 49.40	61.16 ± 45.57	68.12 ± 66.77	ns
SE (%)	81.24 ± 12.26	81.81 ± 11.30	78.38 ± 16.35	ns
NREM1 (% of TST)	6.77 ± 7.06	7.38 ± 7.51	3.75 ± 2.62	ns
NREM2 (% of TST)	52.91 ± 11.52	52.50 ± 11.14	54.93 ± 13.45	ns
NREM3 (% of TST)	18.51 ± 9.52	18.12 ± 9.73	20.43 ± 8.43	ns
REM (% of TST)	21.81 ± 8.08	21.99 ± 7.70	20.90 ± 9.98	ns
Average SpO2 (%)	91.16 ± 9.51	90.60 ± 10.29	94.00 ± 2.20	ns
Minimal SpO2 (%)	80.69 ± 8.03	79.47 ± 7.84	86.76 ± 6.09	<0.001
SpO2 < 90% (%)	13.12 ± 17.76	14.95 ± 18.62	3.99 ± 8.01	<0.05
Average desaturation drop (%)	4.82 ± 2.09	5.11 ± 2.17	3.40 ± 0.52	<0.01
H (n/min)	59.25 ± 9.42	59.43 ± 9.75	58.34 ± 7.73	ns
ESS (points)	9.09 ± 5.13	9.43 ± 5.19	7.63 ± 4.90	ns

BMI, body mass index; HDL cholesterol, high-density cholesterol; LDL, low-density cholesterol, AHI, apnea–hypopnea index; ODI, oxygen desaturation index; TST, total sleep time; SL, sleep latency; REML, REM latency; WASO, wake after sleep onset; SE, sleep efficiency; NREM1 stadium of non-REM sleep 1; NREM2, stadium of non-REM sleep 2; NREM3, stadium of non-REM sleep 3; REM, stadium of REM sleep; SpO_2_ < 90% (%), time with oxygen saturation < 90% (% of total bed time.

**Table 2 life-11-00050-t002:** The carbohydrate metabolism parameters: (A) in the whole study group and within groups A and B, stratified by the presence of OSA.

	Whole Study Group *n* = 102	OSA (Group A) *n* = 85	Without OSA (Group B) *n* = 17	*p* A–B
Glucose (mg/dL)	102.55 ± 14.27	102.53 ± 13.81	102.65 ± 16.85	ns
Insulin (mIU/L)	12.49 ± 10.38	12.29 ± 8.42	13.46 ± 17.50	ns
HOMA-IR	3.30 ± 3.25	3.20 ± 2.45	3.81 ± 5.89	ns
HOMA-IR ≥ 2	64/62.7	54/63.5	10/58.8	ns
IGT	7/6.9	6/7.1	1/5.9	ns
IFG	15/14.7	12/14.1	3/17.6	ns

HOMA-IR, homeostatic model assessment-insulin resistance index; IGT, impaired glucose tolerance; IFG, impaired fasting glucose.

**Table 3 life-11-00050-t003:** The carbohydrate metabolism parameters in groups divided due to criteria of AHI ≥ 15 (group C), AHI < 15 (group D), AHI ≥ 30 (group E) and AHI < 30 (group F).

	AHI ≥ 15 (Group C) *n* = 61	AHI < 15 (Group D) *n* = 41	*p*	AHI ≥ 30 (Group E) *n* = 28	AHI < 30 (Group F) *n* = 74	*p*
Glucose (mg/dL)	104.48 ± 15.24	99.68 ± 12.31	ns	102.54 ± 12.03	102.55 ± 15.10	ns
Insulin (mIU/L)	13.68 ± 9.34	10.71 ± 11.65	<0.05	16.76 ± 11.19	10.87 ± 9.64	<0.01
HOMA-IR	3.63 ± 2.73	2.81 ± 3.88	<0.05	4.41 ± 3.32	2.89 ± 3.14	<0.01
HOMA-IR ≥ 2	42/68.8	22/53.7	<0.05	22/78.6	42/56.8	<0.01
IGT	5/8.2	2/4.9	ns	2/7.1	5/6.8	ns
IFG	8/13.1	7/17.1	ns	5/17.9	10/13.5	ns

HOMA-IR, homeostatic model assessment-insulin resistance index; IGT, impaired glucose tolerance; IFG, impaired fasting glucose.

**Table 4 life-11-00050-t004:** The glycemic metabolism in the study sample divided due to ESS findings.

	ESS ≥ 11 (Group G) *n* = 18	ESS < 11 (Group H) *n* = 25	*p* G–H
Glucose (mg/dL)	97.44 ± 10.45	99.04 ± 10.24	ns
Insulin (mIU/L)	12.31 ± 10.77	12.18 ± 6.53	ns
HOMA-IR	3.03 ± 2.75	3.07 ± 1.78	ns
HOMA-IR ≥ 2	8/44.4	16/64.0	ns
IGT	0/0.0	1/4.0	ns
IFG	4/22.2	6/24.0	ns

ESS, Epworth sleepiness scale; HOMA-IR, homeostatic model assessment-insulin resistance index; IGT, impaired glucose tolerance; IFG, impaired fasting glucose.

**Table 5 life-11-00050-t005:** Results of correlation analysis in the examined group of patients.

	Glucose		Insulin		HOMA-IR	
	r	*p*	r	*p*	r	*p*
AHI	0.15	ns	0.55	0.001	0.54	0.001
ODI	0.24	ns	0.61	0.001	0.61	0.001
TST	−0.13	ns	−0.02	ns	−0.03	ns
SL	−0.16	ns	−0.09	ns	−0.10	ns
REML	0.01	ns	0.24	ns	0.24	ns
WASO	0.18	ns	−0.20	ns	−0.15	ns
SE	−0.07	ns	0.16	ns	0.13	ns
NREM1	0.14	ns	0.54	0.001	0.54	0.001
NREM2	−0.02	ns	0.17	ns	0.16	ns
NREM3	−0.01	ns	−0.43	0.003	−0.42	0.004
REM	−0.12	ns	−0.27	ns	−0.27	ns
Average SpO2	−0.11	ns	−0.24	ns	−0.25	ns
Minimal SpO2	−0.26	ns	−0.29	ns	−0.32	0.032
Spo2 < 90%	0.34	0.027	0.40	0.007	0.44	0.003
Average desat. drop	0.16	ns	0.46	0.002	0.47	0.001
H	0.02	ns	0.18	ns	0.17	ns
ESS	−0.15	ns	0.10	ns	0.06	ns

HOMA-IR, homeostatic model assessment-insulin resistance index; AHI, apnea–hypopnea index; ODI, oxygen desaturation index; TST, total sleep time; SL, sleep latency; REML, REM latency; WASO, wake after sleep onset; SE, sleep efficiency; NREM1 stadium of non-REM sleep 1; NREM2, stadium of non-REM sleep 2; NREM3, stadium of non-REM sleep 3; REM, stadium of REM sleep; SpO2 < 90% (%), time with oxygen saturation < 90% (% of total bedtime; H, heart rate; ESS, Epworth sleepiness scale.

**Table 6 life-11-00050-t006:** Results of regression analysis in the studied group. Model of dependency determining independent predictors of higher glucose concentration.

Model for: Glucose (mg/dL)			
	Intercept	Age (Years)	SpO2 < 90%
Regression coefficient	78.530	0.339	0.154
SEM of Rc	5.562	0.105	0.070
*p* value	<0.001	<0.01	<0.05

**Table 7 life-11-00050-t007:** Results of regression analysis in the studied group of patients. Dependency model determining the independent predictors of higher insulin concentration.

Model for: Insulin (mIU/L)				
	Intercept	ODI	AHI	Age (Years)
Regression coefficient	14.989	0.643	0.412	0.175
SEM of Rc	4.190	0.186	0.168	0.080
*p* value	<0.001	<0.01	<0.05	<0.05

**Table 8 life-11-00050-t008:** Results of regression analysis in the studied group of patients. Model of dependency determining independent predictors of higher HOMA-IR.

Model for: HOMA-IR			
	Intercept	AHI	ODI
Regression coefficient	3.675	0.133	0.199
SEM of Rc	1.180	0.045	0.051
*p* value	<0.01	<0.01	<0.001

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
