# Peer review of "Obstructive Sleep Apnea as a Risk Factor of Insulin Resistance in Nondiabetic Adults"

_life, 2021, doi:10.3390/life11010050_

Round 1

Reviewer 1 Report

Manuscript number: life -1044369, title: “Obstructive sleep apnea as a risk factor of insulin resistance in nondiabetic adults”. The aim of this research was to assess relationship between prevalence and severity of obstructive sleep apnea and insulin resistance among patients with increased risk of OSA without diabetes mellitus. The manuscript is very interesting and it might be accepted in the journal after minor corrections. The discussion is not exhausting and is based only on 2 items of references (26 and 27). I think it should be supported by more studies.

Author Response

Thank you for your opinion, we have already checked all references out and added supplementary references regarding the discussed issue.

Reviewer 2 Report

I have read the article entitled “Obstructive sleep apnea as a risk factor of insulin resistance in nondiabetic adults.” with great interest. The topic is not entirely new, as it is known that OSA is associated with insulin resistance. The novelty lays in the comprehensive evaluation of the PSG, however, before final acceptance the authors should clarify some questions.

Comments:

  • The authors may consider citing https://pubmed.ncbi.nlm.nih.gov/33172084/ which described the prevalence of comorbidities, including diabetes in a large cohort of patients with OSA in Central Europe.
  • How were patients with diabetes excluded? Based on medical history and medications? What about patients with incident diabetes (i.e. glucose levels ≥11.1 mmol/l) found in the current study?
  • Please clarify severe lung disease.
  • Do you have any data on diet and alcohol intake?
  • Please, add data on smoking.
  • Table 1. Please put units to the missing data (AHI, TST, etc.).
  • Table 1. Please explain acronyms.
  • Please add the name of equipment and manufacturer.
  • Statistical analysis. Please provide power calculations.
  • Statistical analysis. The OSA and control groups were different in age and gender. Therefore, comparisons should be adjusted for these factors. Hence, instead of t-tests and Mann-Whitney tests, the authors should perform ANCOVA or logistic regression adjusted on these factors.
  • Table 1. I am very surprised that no difference in sleep architecture was found between OSA and control groups. Please, comment.
  • 1st and 2nd paragraphs. These are repetitions of Table 1. I suggest omitting it.
  • Table 3. I believe it is more appropriate to use ESS≥11 as a cut off for excessive daytime sleepiness. Please, look at the literature.
  • Table 3. Why are there only 22 and 21 subjects? This analysis is definitely underpowered.
  • Results and Discussion. Relationship between REM-dependent OSA and insulin resistance have been previously published in literature. The authors should have data on AHI in REM and nREM separately. Please, perform a correlation analysis and comment on it in Discussion by citing the relevant literature.
  • In general, I found a few grammatical mistakes. The authors should seek assistance from a native speaker.

Author Response

Thank you, we have already revised the manuscript including your suggestions.

Round 2

Reviewer 2 Report

I am happy with the changes and suggest acceptance